# The fisheries governance tool: A practical and accessible approach to evaluating management systems

Jill H. Swasey[1¤a]*, Suzanne Iudicello[2], Graeme Parkes[1]*, Robert Trumble[1], Kara Stevens[3], Martha Silver[3], Cheri A. Recchia[3¤b]

1 MRAG Americas, Inc., Danvers, St. Petersburg, Florida, United States of America, 2 Iudicello Environmental Consulting, Rapid City, South Dakota, United States of America, 3 Strategy, Learning and Evaluation Department, Walton Family Foundation, Washington, DC, United States of America

¤a Current address: Aquaculture Stewardship Council, Hampton Falls, New Hampshire, United States of America
¤b Current address: Mesoamerican Reef Program, Summit Foundation, Washington, District of Columbia, United States of America
* graeme.parkes@mragamericas.com

**Data Availability Statement:** The data used in study and preparation of this tool are all from publicly available sources, which are all referenced or linked in the tool itself.

## Abstract

The Fisheries Governance Tool (FGT) is a comprehensive, evidence-based diagnostic tool that fishery managers, environmental organizations, funders, investors, and other key stakeholders can use to track progress against sustainability goals, identify gaps and challenges that impede continued improvement, and set targets for improvement. The diagnostic tool was developed following a thorough review of existing evaluation and assessment schemes and builds upon many of the credible and widely accepted guidelines and assessment tools currently available. It is built on the premise that the most comprehensive and informative measures of country or regional performance requires evaluation of evidence across three components: 1) the laws and policies governing fisheries, 2) the capacity to implement those policies, and 3) the functioning and performance of the fishery management system and fisheries. The Tool's reliance on empirical evidence allows for an objective, repeatable and rigorous evaluation. Driving this work has been recognition of the importance of identifying and strengthening the enabling conditions for good fisheries management. The FGT offers a unique integrated evaluation of enabling factors and outcomes across the triple bottom line of ecological sustainability, economic efficiency and social/community well-being, with measures spanning a range of identifiable performance levels. Measures identify the building blocks of sound and durable management that lead to more sustainable and responsible fisheries. The Tool was refined through consultation with experts from around the world. The public version of the FGT can be downloaded and allows users to enter data and visualize the results, providing a diagnosis of their management system. The Tool is available in several languages.

**Funding:** The authors did not receive "grants" for this work. Three authors (KS, MS, CR) were employees of the Walton Family Foundation. Three authors (JS, GP, BT) were employees of MRAG mericas, Inc. (MRAG Americas). One author (SI) was a contractor to MRAG Americas. The "funding" structure was a contract between a foundation and a consulting firm. Commercial affiliations, and the role of funders role are described below. The work, of which the submitted article was a minor part, was supported by the Walton Family Foundation through a contract to MRAG Americas Fisheries Technical Division. MRAG Americas is a private consulting and auditing company focused on activities that support the conservation of marine and freshwater ecosystems. MRAG Americas' Fisheries Technical Division conducts analyses of fisheries policy, management, and performance for a variety of clients including environmental foundations, non-government environmental organizations, state, federal and international management agencies, fishery management councils, and industry groups. The contract to create a rubric for scoring national fisheries management policy was awarded to MRAG Americas by the Walton Family Foundation Strategy, Learning and Evaluation Division, originally in 2016 with extension years to 2020 (contract #s C2016-403, C2017-1633, C2019-2857 and 00102041). A subcontract was awarded to Iudicello by MRAG Americas in 2016 with extension years to 2020. The scope of work included research on existing performance assessment schemes, identification of best practices, development of the analytic framework for a scoring tool, workshopping and testing the tool with outside experts, and creating an online, no-cost, downloadable tool. In addition, the deliverables included two rounds of completed scoring of five countries, and submission of an article about the development and application of the tool to a peer reviewed journal. All deliverables were completed. The contract ended in December, 2020. The funder provided support in the form of salaries for authors CR, MS and KS. The funder provided terms of reference for the study, as outlined in the manuscript, and participated in the selection of case study countries, but did not have any additional role in the data collection and analysis, decision to publish, or preparation of the manuscript. The specific roles of these authors are articulated in the author contributions section. At the onset of the contract, co-authors CR and MS were on the staff of Walton's Strategy, Learning and Evaluation division. In 2018, co-author KS joined the Walton staff. At the onset of the contract, corresponding author GP, and co-authors JS and

# 1 Introduction

Fish and fisheries play critical roles in ocean health, community well-being, food security and economic stability, and ensuring their sustainability is a key goal of governments, civil society, philanthropies and development institutions. Development assistance and philanthropic grants to fund marine related initiatives totalled over US$1 billion in 2016. US$206 million of this was directed to marine fisheries by the philanthropy sector [1]. Collectively, this work spans an enormous variety in types of fisheries, affected species, habitats, ecosystems, jurisdictional and policy contexts, management capabilities, and socio-economic conditions.

Among the increasing number of philanthropies focused on improving marine fisheries sustainability, the Walton Family Foundation (WFF) has long recognised of the importance of sound and durable fisheries management. In pursuit of this there is a need to identify and strengthen the enabling conditions that support successful interventions. A robust definition of success in this work is context-specific, considering, for example, local socio-economic goals and agreement on desired environmental states. Nevertheless, common requisites for and indicators of long-term sustainability can be identified, including sound management policies, sufficient institutional capacity and resources to implement policy and practical management measures and tools that keep environmental impacts within acceptable bounds.

In countries where they worked, WFF was seeking to assess progress against goals set to reflect local circumstances, without the pass-fail value judgements that arise in benchmarking or certification programs. Understanding the wide array of contextual factors that contribute to effective fisheries management systems is key in this process. These contextual factors range from the basic principles of good governance (e.g., goal-setting, transparency, accountability, public participation), to those of good management (e.g., quality, availability and use of science), to myriad other considerations appropriate to individual fisheries (e.g., co-management mechanisms). This led to the development of the novel Fisheries Governance Tool (FGT) presented in this paper. The FGT directs users in the building of an objective and comprehensive country-level evaluation across an entire fisheries management system, an analysis of the gaps in that system, and a means of measuring progress over time relative to locally established goals as changes are made and gaps are filled. Where country is not the most appropriate jurisdictional level at which to make the evaluation, the FGT can be applied equally well at a regional, state or district level.

While its origin was in a specific evaluation requirement, we believe the resulting tool has much broader utility and applicability for a wide range of stakeholders, spanning philanthropic organizations, development agencies and governments; hence the motivation for preparing this manuscript for publication in a peer-reviewed open access scientific journal to put the FGT before a technical audience in the hope that this will encourage its boarder uptake and further development.

In the following section we review briefly the measurement of sustainability performance in fisheries to provide additional context for the genesis of the FGT. In the remainder of this paper we describe the FGT, the process of indicator development, including testing, review and revision, and introduce the first iteration of the software package built for its implementation which can be freely downloaded and used. We also discuss initial results and how we hope the tool can be developed further in the future.

## 2 Measuring sustainability performance in fisheries

While it is a novel evaluation tool, the FGT draws from existing guidelines, standards and evaluation mechanisms, many of which can trace their origins back to The FAO Code of Conduct for Responsible Fisheries (CCRF). Created in 1995 by the Food and Agriculture Organization

RT were employees of MRAG Americas. Billable hours related to the project were charged to the Walton Family Foundation contract. The author contributions section of the submission describes the roles of each of these co-authors. Co-author SI is an independent researcher, writer and consultant, and sole proprietor of Iudicello Environmental Consulting. A sub-contract with MRAG Americas from the onset of the Walton project was one of more than 10 separate clients/ projects during the period 2016-2020. Billable hours related to the project were charged to the MRAG Americas sub-contract. The author contributions section of the submission describes the role of this co-author.

**Competing interests:** The authors have no competing interests. The work conducted by MRAG Americas and by Iudicello did not interfere with, nor could be perceived as interfering with the development of the Fisheries Governance Tool or the authorship of the article describing the project the full and objective presentation, peer review, editorial decision-making, or publication of research or nonresearch articles submitted to one of the journals. The corresponding author confirms that affiliation with MRAG Americas and Iudicello Environmental Consulting does not alter our adherence to PLOS ONE policies on sharing data and materials.

(FAO) of the United Nations, the CCRF and the four International Plans of Action (IPOAs) established under it [2–4] provide the normative guidance on sustainable fisheries management. The CCRF does not set out specific performance indicators, but articulates established principles and international standards of behavior for responsible practices with a view to ensuring the effective conservation, management and development of living aquatic resources, consistent with the triple bottom line of ecological sustainability, economic efficiency and social/community well-being. Alongside several normative legal instruments for fishery management (1982 UN Convention on the Law of the Sea (UNCLOS), the 1993 FAO Compliance Agreement (on effective control by the flag State over fishing vessels), the 1995 UN Fish Stocks Agreement (UNFSA; relates to straddling fish stocks and highly migratory fish stocks), and the 2009 FAO Agreement on Port State Measures), the CCRF and the IPOAs formed a key element of the criteria established for the performance reviews undertaken by 19 Regional Fisheries Bodies (RFBs) between 2005 and 2014 [5]. The RFB reviews generally covered four key topics: Conservation and management of fish stocks; Compliance with and enforcement of international obligations; Legal framework, financial affairs, organization; and Cooperation with other international organizations and non-member States. After several RFB reviews, the "Kobe criteria" [6] were developed during the first Kobe meeting of tuna Regional Fishery Management Organizations (RFMOs) to consolidate review criteria, also adding a fifth criterion on Financial and administrative issues. Several non-tuna RFBs also adopted the Kobe criteria, but they were often modified to fit the requirements of the RFBs.

Indicators of sustainability and responsible practice cover both inputs (e.g. management attributes and other enabling factors) and outputs (e.g. stock status and other environmental impacts) to varying degrees. Evaluations tap a range of information sources including existing data, monitoring programs, interviews and expert judgment. The high cost of bespoke monitoring programs result in assessment of outputs relying mainly on data collected routinely for other purposes, with consequently variable levels of success. Data characterizing enabling factors such as specific management attributes and interventions may be more readily obtainable, but establishing causal relationships between those inputs and specific outcomes at the national, regional, local, fishery and/or stock levels is difficult, often confounded by spatially and/or temporally overlapping effects. Nevertheless [7], demonstrates the value of improved fishery management, showing that countries with intensive management and quantitative stock assessment generally have stock status trending towards or at target levels, while countries without intensive management and quantitative stock assessment tend towards fishery status below target levels.

Building on the principles established in the CCRF and related texts, the fishing industry, NGOs, national governments, and international organizations have since contributed to a diverse array of options for evaluating fishery management performance and identifying responsible sourcing for seafood products. Independent schemes that evaluate individual fisheries include voluntary certification programs such as the Marine Stewardship Council (MSC) [8], Alaska Responsible Fisheries Management (RFM) [9], and Friend of the Sea (FOS) [10]; and seafood ratings such as Monterey Bay Aquarium's Seafood Watch Program and Sustainable Fisheries Partnership's (SFP) FishSource. To date, these schemes have focused mainly on the environmental impacts of fishing, with less direct attention given to social/community well-being, economic efficiency and governance performance. They also generally describe higher, more desirable states to distinguish sustainable practices. The MSC Standard [8] for example, functions cumulatively with its 60-80-100 scoring guideposts of each scoring issue being either met or not met. There is no description of performance at lower levels with any scores below "60" causing a fishery to fail certification. Lower-level states have been described elsewhere to demonstrate the pathway towards sustainability (e.g. [11]) and fisheries engaged

in Fishery Improvement Projects (FIPs, see https://fisheryprogress.org/) focus on incremental changes that will ultimately meet the higher standard to achieve certification.

In an effort to help the seafood industry identify which certification schemes are properly aligned with the CCRF and accompanying certification guidelines [12–14], the Global Seafood Sustainability Initiative (GSSI) has developed a Global Benchmark Tool [15]. While not a fisheries evaluation or certification scheme, the GSSI Benchmark lays out the fundamental aspects of responsible fisheries management identified by FAO that robust certification schemes must include.

Rather than offering a certification, the Seafood Watch Program provides independent ratings of a range of fisheries around the world that sell seafood products in the US. Similarly, SFP's FishSource compiles and summarizes species level information on fishery science and management for use by major seafood buyers. Both ratings systems summarize publicly available scientific and technical information for broader audiences. Other recommendation schemes (e.g., those of the Environmental Defense Fund and the Safina Center) use Seafood Watch information to rate seafood products, but may reach different conclusions regarding sustainability due to different weighting of indicators.

In a more comprehensive evaluation framework, the Fisheries Performance Indicators (FPIs) of Anderson et al. [16] present a detailed series of metrics covering the triple bottom line across output and input dimensions, each scored on a coded scale from 1 to 5. For the output metrics, higher scores reflect better performance, with levels capturing (projected) quintiles or key performance benchmarks across global fisheries and 3 being a level below which improvement could be considered. For input metrics, the level descriptors are monotonic, with no presumption that higher output scores result from higher input scores. To conduct a comparative assessment across multiple countries, Melnychuk et al. [17] used expert surveys of 28 major fishing nations to assess 13 country-level fisheries management attributes covering four dimensions (research, management, enforcement, socioeconomics) on four stock status criteria. Both of these studies used expert assessment of one form or another, in part to facilitate application to data poor fisheries and sectors, but principally to enable assessment and comparison across a large number of entities (fisheries, species, countries etc.) as rapidly as possible.

All of these various instruments, tools and guides and their contribution to the current understanding of how fisheries can be more responsible and sustainable were reviewed as part of WFF's consideration of how best to make its assessment of progress in grantee countries. Systems and procedures for fisheries governance are changed incrementally, with periodic wholesale revisions and updates of fisheries policy, laws and management strategies in the expectation of improved outcomes. Yet, governments and stakeholders rarely have the necessary evaluation tools and information to make their own accurate and repeatable assessments of progress, making it harder to advance, or advocate for, the requisite interventions that will enable and support an effective transition toward more sustainably managed fisheries. To change this dynamic, there was a preference for a governance-based evaluation, applicable at the county or regional level, with metrics covering he full range of performance, describing basic to advanced-level conditions in a cumulative structure, such that each metric could be assessed as either present or not. There was a particular interest in recognizing the importance of a sound and durable foundation of essential governance structures and institutional capacity on which to base the higher level performance associated with sustainable fisheries. While the states articulated in the metrics should conform to international norms, particularly at the more advance levels, there was a desire to avoid any particular external standard or benchmark driving the evaluation outcome. Instead, the tool needed to accommodate locally set goals and aspirations, providing in-country stakeholders with the means to make their own robust and

repeatable assessments. Early research showed that to meet these conditions required a bespoke evaluation tool built from the ground up.

## 3 Materials and methods

### 3.1 Outline of the FGT

The FGT explores fisheries management status and performance relative to local objectives through the intersection of three key components: 1) fisheries management policy, 2) the capacity to sustain and implement that policy, and 3) management measures and tools that advance the achievement of the goals and objectives of that policy. This recognizes fisheries management as a complex activity including goal setting, planning, information gathering, scientific analysis, consultation, decision-making, allocation of resources, and design and implementation of strategies and activities. While the tool measures and acknowledges low levels of "basic" performance, an assumed pre-requisite is a functioning level of government and civil society such that the essential tenets of governance apply, including a recognized authority to articulate and implement policy. Building up from this, the Policy component (1) is the set of basic principles and associated guidelines, formulated and enforced by the governing body of an organization, to direct and limit its actions in pursuit of long-term goals. The Capacity component (2) is the ability of people, organizations and society as a whole to manage their affairs in a way to successfully implement their policies, and plans and thereby attain their goals. In essence, this is about who does the science, management, administration and decision making, how they do it, and how well this reflects and supports the achievement of the policy objectives. Component 3 is the actual measures, strategies and plans designed and implemented to achieve the policy objectives, including the rules governing fishing activities and their enforcement.

An assessment using the FGT relies on documentary evidence, graded for quality, to score performance across these three components according to series of over 200 measures, organized into Performance Areas and Indicators. The evaluation process is comprehensive, can be scaled at national, regional, local and fishery levels, and gives users flexibility to set their own goals and measure progress against them. The FGT does not start by asking "what data are available?" Instead, the FGT has a comprehensive set of indicators to answer key questions, regardless of whether the necessary data are currently collected. Aside from the performance assessment, the results therefore also serve as gap analysis showing where data are lacking or of poor quality and which aspects of fishery management and operations require additional monitoring. This provides an evidence-based approach that allows for an objective and repeatable evaluation with scientific rigor.

The FGT enables users to assess policies that are on the books, but also evaluate whether they are implemented with the required resources. This avoids incorrect attribution of fishery-level outcomes to specific policies or actions. Information on an individual fishery's functioning is important, but may not be representative of overall system performance. The FGT can identify where there is a sound management system in place leading to good outcomes across a range of fisheries, as compared to an apparently well-performing fishery that is a one-off, operating under a management system that lacks longer-term stability and durability.

The structure of the Tool emphasizes actions that are considered fundamental to enduring success. For example, an advanced action taken in a pilot project or narrowly targeted fishery carries less weight than a broadly applied basic measure such as catch limits. This weighting provides a means to look at diverse fisheries using a consistent frame.

Over a four-year period, the FGT evolved from a set of key questions and indicators into a sophisticated, comprehensive, flexible and user-friendly diagnostic tool. This evolution

included rigorous review, testing, feedback, revision, and final pilot runs across three main phases: (i) indicator development and internal review; (ii) testing, external review and revisions; and (iii) design and roll-out of a downloadable tool. We describe these phases in the following sections, with the technical details of the FGT inserted between the descriptions of phases (ii) and (iii) including data needs, scoring methodology and the effects of data quality.

## 3.2 Indicator development and internal review

The four-person development team at MRAG Americas had extensive experience developing and using assessment systems created by governments, NGOs, and private entities. The team was supported by WFF's Strategy, Learning and Evaluation professionals. Review of the literature, practice, and results of fisheries performance evaluations around the world led to the decision to develop indicators encompassing ecological, economic, and social considerations for both enabling factors and performance outcomes. Indicators and measures were informed by the CCRF and built on a review of and experience in multiple fisheries performance evaluations (Section 1).

The Tool comprises a series of Performance Areas, Indicators and Measures to answer three key questions aligned with the Components of the tool: Policy, Capacity, and Performance (Fig 1).

1. Policy: Does the fisheries policy provide the basis for rational and effective governance and management of the nation's domestic fisheries, and its orderly and legitimate participation in international fisheries?

2. Capacity: Does the nation [or other administrative entity] have the capacity to reliably and consistently implement the fisheries policy in successful pursuit of the goals articulated therein?

3. Performance: Does fisheries management function in a way that effectively and efficiently implements the fisheries policy?

Performance Areas are used to categorize the Indicators that show the level at which the system is performing based on the Measures achieved. The Measures themselves are descriptions of features that are either present or not in a fishery governance and management system. For example, an Indicator in Component 1 that national policy promotes sustainability within fishing communities includes Measures such as goals to optimize access, to promote safety of life at sea, to recognize customary and aboriginal rights, and similar metrics drawn from other established standards. The measures capture a full range of practice and outcomes, building up from the most basic to current best practices, but there is no external stipulation of what levels should be achieved within this range for each Indicator and Performance Area.

## 3.3 Testing, external review and revision

Using the first draft of the FGT, the development team worked with in-country consultants to conduct pilot assessments for a selection of test countries. This exercise revealed challenges in acquiring the necessary information, inconsistencies in how users assessed the presence of Measures, and uneven documentation of the evidence base. Feedback identified variations in how users approached the assessments, and some conflicts in how experts, grantees, and/or country officials viewed the results. Although users agreed that the Tool covered all the key parts of what they wanted to know, its complexity made visual presentation of the results challenging.

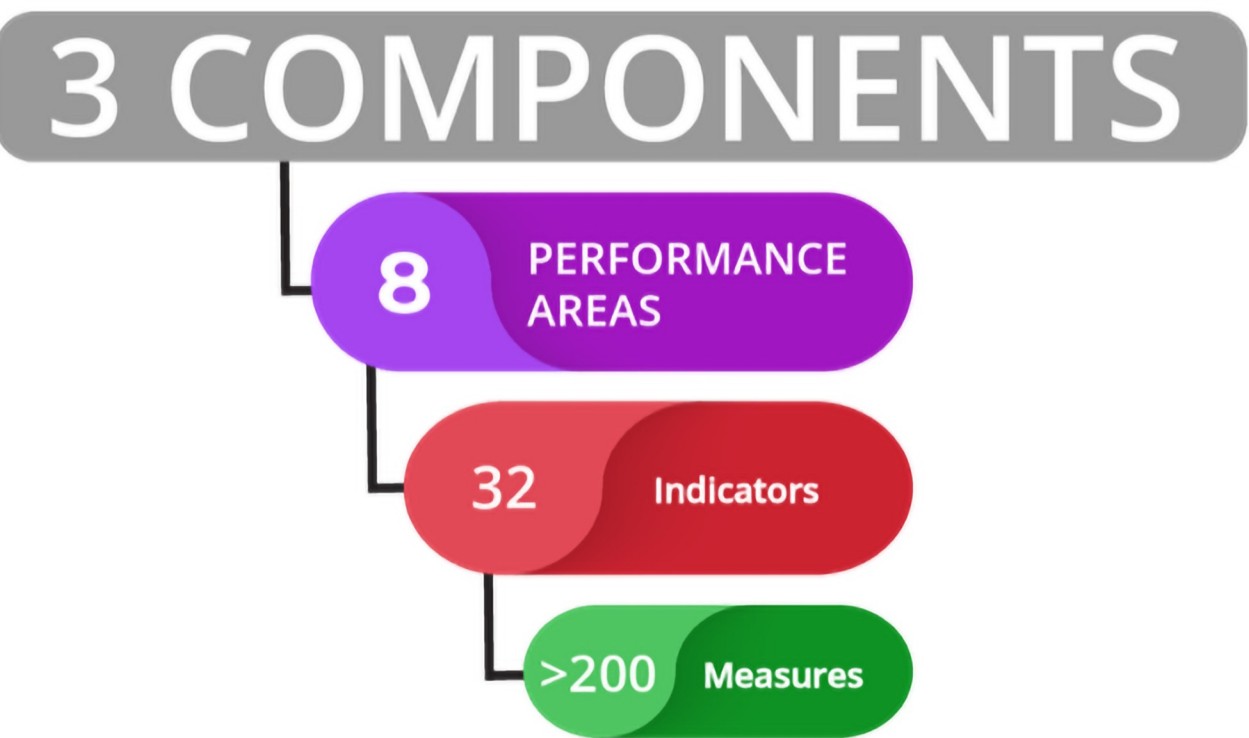

**Fig 1. The hierarchal structure of the fisheries governance tool begins with high-level components, that comprise eight performance areas, under which 32 indicators and over 200 measures collectively capture a comprehensive picture of policy, capacity, and performance outcomes.**

Resulting revisions improved navigation of the Tool, clarified intent and terminology used and created more detailed guidance for evaluation, evidence and documentation, data acquisition and data quality. In November 2018, a peer-review working group of eight experts, representing government fishery managers in developed and developing nations, universities, and NGOs, was convened to undertake an external review. The technical working group was tasked with assessing the feasibility, applicability and completeness of the Tool, and identifying constraints that might hinder its implementation, including data requirements, documentation, and accessibility for developed and developing countries.

The external reviewers first identified the potential value of the FGT as a more broadly applicable diagnostic tool. Based on this advice, the development team set about expanding the scope and application of the Tool making it accessible to external stakeholders and other users. The reviewers also urged the team to revise the functionality of the framework to demonstrate that as evaluations are repeated, the score trajectory itself has value. Among other significant changes that emerged from the technical review was an expansion of what was meant by "governance" in Component 1, clarifying its applicability not only to multiple levels of governance (national, regional, local, and fishery specific), but also to non-state governance.

A major revision followed, with clarification of terminology, labeling the major segments of the Tool, expanding metrics in social and community indicators, adding Measures on adaptation, incentives, perverse incentives, trade-offs, and ensuring that transparency was evaluated in all decision processes, from policy through capacity and at the fishery management level. Other enhancements included coherent messaging, graphic design, and data visualization to help users access and operate the Tool.

### 3.4 Technical details

**3.4.1 Data needs.** An evaluation using the FGT is largely a desk-based activity, but requires access to first-hand knowledge of the country/management system in question, verified through source documents and relevant contacts. Acquisition of reliable evidence is fundamental for evaluating whether or not a given Measure is met (see Section 3.4.2). In some instances, detailed information may be available from verified online sources, but for many countries it isn't, in which case the user will need to acquire evidence and documentation directly through available sources, such as agency publications. Evaluations may also use credible secondary sources that compile and summarize information on fishery performance such as MSC certifications, Seafood Watch reports, FishSource profiles, etc., but these should not be regarded as primary references.

Wherever possible, evidence that enabling factors are present (across Components 1 and 2) should be drawn from primary documentation, such as laws, regulations, decrees, and policy statements. Because the Tool can be applied to a range of governance scales this may require documentation from one or several levels of government potentially ranging from international, country-wide, to regional or local. Evidence may also come from secondary and summary reports, but the evaluation results will carry less weight, because the evidence is of less than primary quality. Qualitative and anecdotal evidence directly from country and fishery experts is regarded as the lowest level of information on which a determination can be made (see Section 3.4.3 regarding how the Tool addresses data quality).

Component 3 requires documentary evidence of management system attributes and outcomes such as harvest control rules and stock assessments. This represents the management strategies, regulations and performance achieved across the triple bottom line for individual fisheries. Triple bottom line performance outcomes can be measured at various scales (Table 1). Selected fisheries that operate within the jurisdiction of the relevant governance body serve as proxies of the functioning of the overall management system. We rely on this approach to provide a representative sample of fishery outcomes, without the burden of assessing all fisheries.

Evidence is used to evaluate each Measure in the FGT on a three-point scale: fully met ("Yes"), partially met ("In Part"), or not met ("No"). A Measure may also be not evaluated ("Not Evaluated"), because no data were available to make a determination regarding the extent to which it was met; this differs from "No", which requires specific evidence that the Measure is not met. Measures under Component 3 may be recorded as "Not Applicable" if a particular management strategy, for example, does not apply to the characteristics of a fishery being assessed (this option is not available for Components 1 and 2). Tables provided as supporting information detail the standards of evidence and examples for assessing the extent to which a Measure is met under Components 1, 2 and 3 of the FGT.

**3.4.2 Scoring.** The Measures are graded on a four-point scale: Basic; Adequate; Good; and Better. Each Measure can be met, or not, independently of the others. They are not performance grades of the same Measure, as seen for example in the 60-80-100 guideposts of the MSC standard (where the 80 level cannot be met without first meeting the 60 etc.). Nevertheless, Measures within a single Indicator are related (in that they all contribute to the Indicator) and designed to be cumulative in a conceptual sense: the Basic and Adequate Measures are the essential foundations for establishing sound and durable fisheries management, while the Good and Better Measures represent more sustainable and responsible management. It is possible for systems to have Good and Better Measures in the absence of Basic or Adequate Measures, but the expected benefits of the Good and Better Measures may be undermined if the Basic and Adequate Measures within the same Indicator are not also met (an effect that may

**Table 1. Meaningful scales of measuring performance outcomes.**

| | | Scale | | | | | |
|---|---|---|---|---|---|---|---|
| | | **National** | **Regional** | **Local** | **Fishery** | **Stock** | **Environment** |
| **Triple Bottom Line** | **Economic** | ✓ | ✓ | ✓ | ✓ | | |
| | **Community** | | ✓ | ✓ | ✓ | | |
| | **Ecological** | | | | ✓ | ✓ | ✓ |

also cross over between Indicators). This is because the necessary underpinning structures and processes for enduring good performance are not in place; Table 2 provides an example of the grades of selected Measures within the Indicator of Principal Elements under Component 1.

The FGT generates a numerical score based on the grades of the Measures met and not met. This score was developed principally to facilitate compilation and illustration of the assessment results in a user-friendly way. The essential value of this is as a means of tracking of progress over time: as more Measures are met, so the numerical value increases. For this purpose, a simple starting point would be to score 1 for every Measure fully met, 0 if not met and 0.5 if partially met. However, this gives the impression that every Measure (Basic; Adequate; Good; or Better) has the same incremental contribution to the building of a successful fishery management system. In practice this is not credible, both in terms of their cumulative contribution within Indicators, and also across Indicators, Performance Areas and Components. It would be a useful elaboration of the FGT to develop a scoring model that plausibly reflects the relative incremental contribution of the Measures, and given time in practice, and accumulation of significant quantities of time series data, this might be possible. At this early stage we take a first step and postulate that the Basic and Adequate Measures will have a greater incremental effect on performance improvement than the Good and Better Measures. Even though the Basic and Adequate Measures may not reach what is typically regarded as a target level of performance (e.g., the 80 level in an MSC assessment), this step up from a base of essentially no management measures will have a significant incremental benefit. As the system becomes more sophisticated with the introduction of the Good and Better Measures, performance continues to improve, but there is a diminishing return. Fig 2 plots this principle as a curve, showing how for the same step along the x-axis (effort in implementing) the relative degree of improvement achieved (the y-axis) is progressively less. The actual relationship between effort and improvement (and by inference between the Measures and performance) is obviously highly complex, variable and not conveniently shaped as per Fig 2. Nevertheless, we used this generalization to develop the preliminary scoring system adopted in the FGT. We allocated

**Table 2. An extract from Component 1 of the FGT showing measures within a single indicator ranging from basic to better.** Whether the system meets each measure is assessed using the available data.

| Performance Areas | Indicators | Measure | Grade |
|---|---|---|---|
| 1.1 Policy Content | 1.1.1 Principal Elements | 1.1.1.1 An identifiable fisheries management policy exists. It is generally applicable and is recognized internally and externally as the policy that guides fisheries management at the country, regional, and local levels. | Basic |
| 1.1 Policy Content | 1.1.1 Principal Elements | 1.1.1.2 The fisheries management policy contains the principal elements of a functional policy; it is clearly thought out, with specific goals to guide management strategies that the state and legitimate interested parties have agreed will provide optimal benefits in the long term. | Adequate |
| 1.1 Policy Content | 1.1.1 Principal Elements | 1.1.1.6 Clear long-term objectives that guide decision-making, consistent with the specific ecological, economic, and social goals, are explicit within management policy. | Good |
| 1.1 Policy Content | 1.1.1 Principal Elements | 1.1.1.11 The policy mandates clear long-term objectives for fisheries management throughout the management system. | Better |

**Fig 2. A stylized fisheries improvement curve showing the grades basic, adequate, good and better for measures in the FGT; the greatest improvement per unit of effort is at the lower end of the curve.** In this case "effort" is a nominal measure that in practice will vary between indicators, but in some instances may simply equate to cost (e.g., in the implementation of more sophisticated monitoring systems).

nominal numerical scores for Measures fully met as follows: +4 for Basic, +3 for Adequate, +2 for Good and +1 for Better. For Measures partially met, these scores were halved (a Basic measure achieved In Part received a score of +2, for example). While this is likely to be closer to reality than a simple score of unity for each Measure fully met, it does not address the many complexities, including the differential contributions of Measures at the same level (i.e. not all Basic Measures will have the same incremental benefit). Whether or not a detailed examination of this issue would yield a more elaborate scoring mechanism that significantly enhances the utility of the FGT remains to be seen.

This allocation of nominal scores to the Measures enables calculation of an overall score at various levels (Indicator, Performance Area and Component) according to which are met, partially met, and not met. A simplified illustrative calculation is provided (Fig 3); dividing the total score achieved by the total possible score gives a percent score achieved. In doing this, we note that while allocating numerical values to a categorical scale of text descriptions (i.e., the Measures) may be expeditious and help to summarize results, it is open to misinterpretation. We therefore caution users against ascribing more meaning to the scores than they can properly support.

**3.4.3 Supporting evidence.** Recognizing that performance evaluations will be performed by individuals or groups with varied expertise, and that individuals bring their own level of knowledge and perspective to the task, the FGT includes descriptions of the types and

---

**Calculation Example:**

- Measure A (Basic) – Yes = 4 points of 4 possible

- Measure B (Better) – In part = .5 points of 1 possible

- Measure C (Adequate) – Yes = 3 points of 3 possible

- Measure D (Good) – No = 0 points of 2 possible

*Sum of Measures Scored (7.5)* **divided by** *Total Possible Score (10)* **=** *Score Achieved %*

*(75%)*

Where a Measure is "Not Evaluated" or "Not Applicable", the Measure is removed from the total

possible score.

---

**Fig 3. Simplified illustrative calculation of an overall score based on measures scored.**

standards of evidence required to achieve a "Yes" or "In Part" for a Measure (see Supporting information). Users are encouraged to add narrative explanations of the evidence supporting their conclusions to promote consistency across repeated assessments.

The FGT incorporates a data quality scale, ranging from best data to no data (Table 3). Similar data quality indices have been applied elsewhere; for example, see the index employed in Productivity and Susceptibility Analysis methods of Patrick et al. [18]. Scoring the quality of the data provides an additional lens through which the results should be viewed, including for example, where an assessment showing a system is performing well is based on outdated information. Progress over time may also be reflected in improved data quality scores from one assessment to the next.

**Table 3. Data quality scale applied to the assessment of each measure in the FGT.**

| Tier | Description |
|------|-------------|
| 1 | **Best data**. Referenced, agency document, peer review, published, current within the last two years. |
| 2 | **Good data**. Grey literature, foundation reports, expert interviews, government websites, media articles (triangulation/confirmation), data within 3–10 years. |
| 3 | **Limited data**. Outdated (>10 years), anecdotal, traditional ecological knowledge (triangulation/confirmation). |
| 4 | **No Data**. Measure cannot be evaluated. |

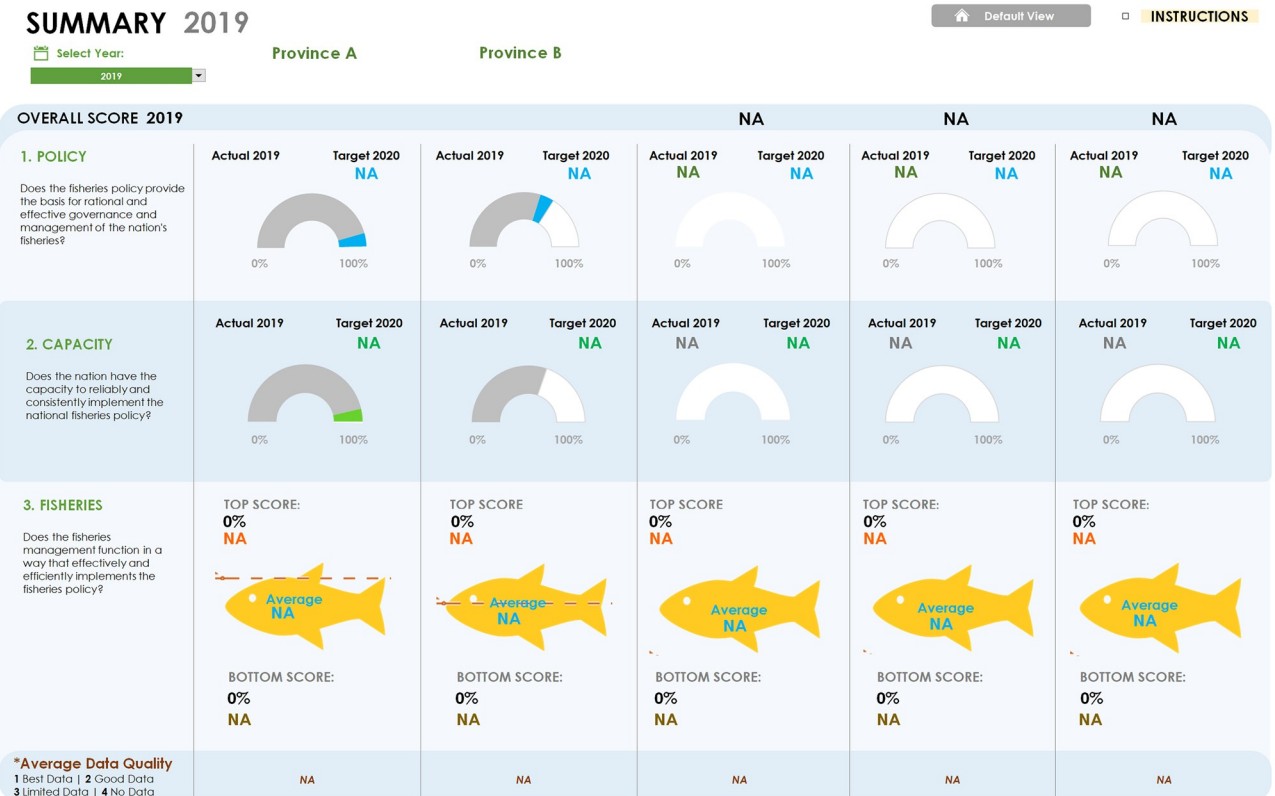

**Fig 4. Outputs for Components 1 and 2 show a country's progress based on measures achieved relative to optional targets set by the user.** Component 3 shows progress of individual fisheries, also against optional targets. Fishery performance can be viewed in the context of the policy and capacity components to investigate where key strengths and weaknesses exist.

### 3.5 Downloadable tool and example outputs

To make the FGT broadly accessible, a user-friendly, downloadable version was built out by a commercial software developer specializing in Excel applications. The public version of the Tool includes enhanced user guidance, including more definition of technical terms and further instruction on the functions of the Tool, but maintains the essential integrity and rigor of the original. The Tool allows for integrated data input, processing and visualization of results. Users are provided flexibility to customize the Tool according to the situation they are assessing (e.g. two fisheries in one state; ten fisheries in one country, etc.). They can set their own targets and baseline against which subsequent performance will be assessed. Essential functions built into the Tool include the ability to sort results, add years of data to track progress, and establish targets across Indicators. The visual outputs summarize results across various levels of assessment, showing the strengths and weaknesses of the system being assessed.

The FGT can be downloaded at www.fishgovtool.com/ in English, Spanish, or Bahasa versions to any device that supports Excel 2013 or newer. The file includes example data that illustrate how the results of a real assessment are displayed, with detailed instructions for how to navigate and start entering real data to conduct an assessment. The download includes instructions for use, public-facing materials to summarize, visualize, and share results of performance evaluations, and links to a website with additional guidance and downloadable resources. To date the Tool has been applied in five countries, monitoring progress over time. Outputs from the evaluation allow visualization of results across Components and Indicators as shown in Figs 4–6.

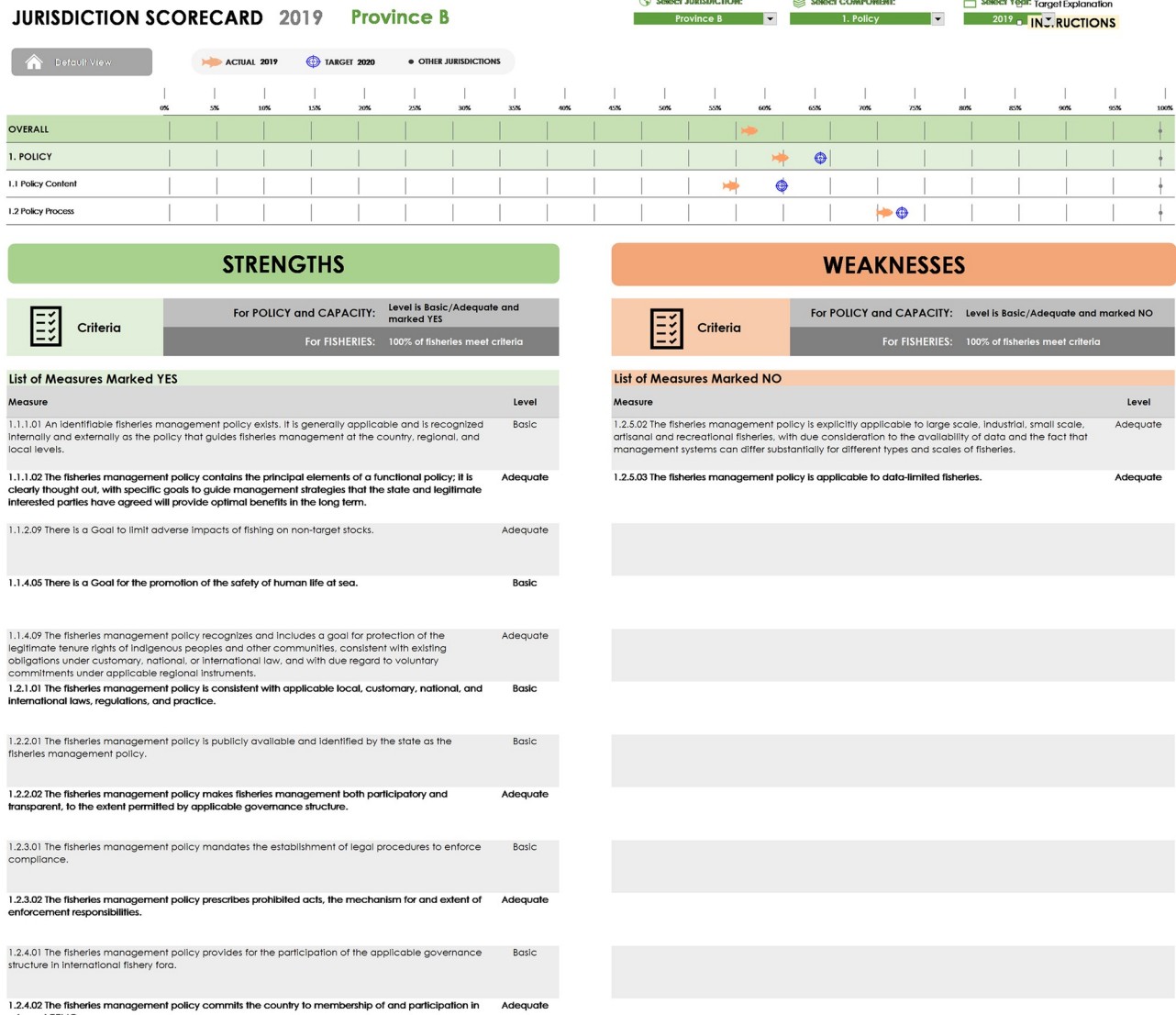

**Fig 5. A deeper dive into the results shows the strengths and weaknesses within a component.** Progress across performance areas illustrates where advances have been made and where work remains to be done, although the latter is specific to the context of the country, province or fishery under review and the optional targets set. Strengths and weaknesses across individual indicators are provided as a diagnostic feature to highlight areas for further review or potential intervention.

## 4 Discussion

Since the launch of the MSC in 1997, approaches to evaluating sustainable fisheries and seafood have emerged from NGOs, governments, retailers, and international organizations. Most are based upon the fundamental tenets of the CCRF, and many have been diligent about updating and improving their standards with transparent stakeholder processes to enable critiques and improvements. The scientific literature is rich with ideas on how to improve assessment of fishery performance, suggesting and testing new methods and pathways to evaluation. The structure of the FGT and its performance diagnosis are based on a sound and detailed understanding of how fisheries management works and the factors that are likely to lead to success, as reflected in the CCRF and other codes, standards and guidelines.

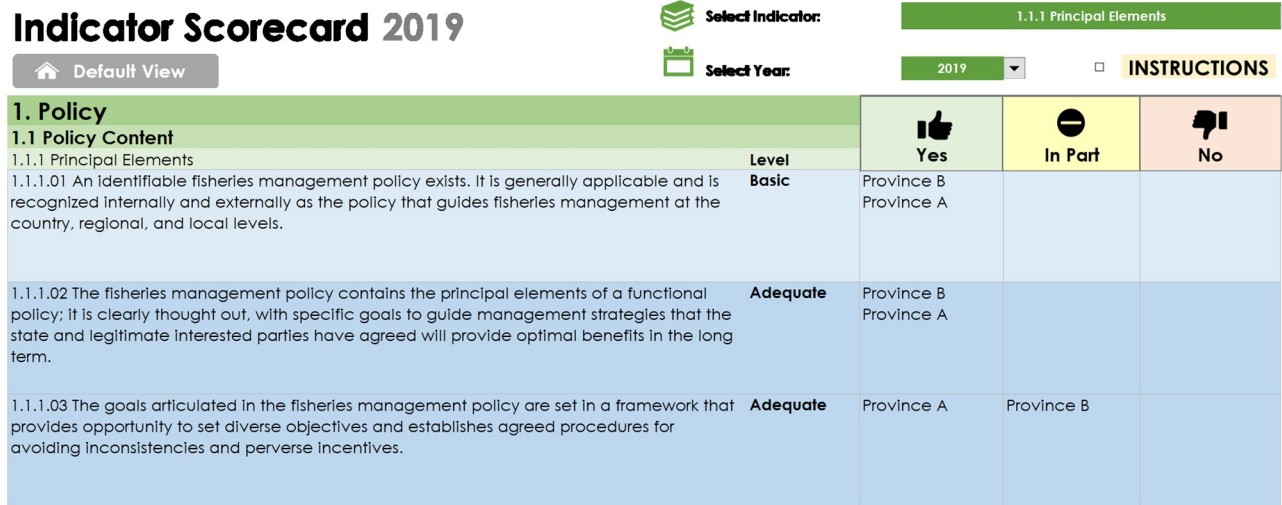

**Fig 6. Reviewing results at a high level provides a general overview of progress, or lack thereof.** The indicator scorecard enables the user to select a performance area of interest and compare where multiple jurisdictions have been assessed. This detailed look across indicators focuses the output on deficiencies in performance and reveals gaps where additional information may be needed to better inform the evaluation.

Originally intended as an internal evaluation tool for WFF country projects, the FGT evolved though analysis of and comparison with prior methods and consultation with scientists creating, publishing and testing new approaches. Throughout the development of the FGT, the development team apprised others working in the field of the project, engaged potential stakeholders via webinars and presentations, and invited ongoing external review. Reviewers and testers pushed the team to find ways to make the tool more applicable in a range of circumstances and diverse management regimes. These challenges resulted in numerous improvements, including:

- A widely available and familiar platform (Excel) that facilitates user inputs and interactive outputs.

- Translation into languages other than English.

- Enhanced information on how to use the Tool, including the time and knowledge required to score and document the evidence for the evaluation consistently.

- Use of expert interviews where data or published documents are not readily available, while maintaining a preference for empirical evidence.

- Expanded guidance and best practices on use of traditional knowledge and customary law.

Not all fisheries are rule-based. The range of possible or already existing systems includes constitutional, legislative, regulatory, collaborative, operational, provincial, local, community and contractual options. In addition to providing users the presence of factors to analyze in rule-based and rights-based systems, the FGT recognizes the need for consideration of small-scale, data-poor and non-state governance systems. The tool includes more than two dozen measures of a system's ability to accommodate assessment of fisheries that are small-scale, data-poor, community-based, use traditional knowledge, or apply tenure rights. During review and testing, the authors had limited opportunities to apply the tool to traditional, collaborative, community-based and other management and decision processes that were not law-based. In Indonesia, where small-scale fisheries in one of the largest EEZs in the world are among top

contributors to wild capture fish landings, the FGT was applied to examine progress in 10 fisheries. These ranged from small-scale handline and gillnet fisheries for pelagics to longline vessels from 1–60 GT fishing for tuna. The dispersed nature of management at district levels (such as data collection, enforcement, and decision-making) required using interviews with in-country experts familiar with the fisheries rather than relying solely on government reports or published data. Similar adjustments to applying the FGT were made in the second round of assessing the performance of 48 fisheries in five nations representing diverse fleets, species, gear, volume, and value. A pilot project to employ the FGT in one or more Pacific Island SIDs was suggested as a way to confirm the applicability of the tool where community-based and traditional management approaches are used—solely or in conjunction with rule-based or access agreement measures—to manage marine resources. Arrangements for such a study were not yet made at the time of this writing.

The FGT puts power in the hands of managing agencies, environmental organizations, funders/investors, and other stakeholders to designate desired outcomes and objectively assess their systems. With the Tool, users can establish a robust baseline and measure genuine progress against clear metrics over time and identify information and performance gaps and other challenges that impede continued improvement. Importantly, managing agencies can see their progress relative to the objectives set in their country's own policy and management plans, rather than against an external standard that may not be relevant or realistic in a specific context.

While converting the FGT results into a numeric score is useful for presenting results and tracking progress, it risks the outputs being misinterpreted or used in ways that are not intended. The FGT calculates a percent of the theoretical maximum score using simple mathematics. If this percent increases over time as the assessment is repeated, then this should be a good sign. Also, a closer look at the results can show where specifically the score can be improved, and particularly what adjustments to the system (e.g., changes in policy, budgets, enforcement capacity, management strategies, etc.) should result in the greatest incremental change. However, given the limited basis for the scoring at present, we caution against this number being used to infer, for example, that a country is X percent of the way towards achieving some notion of "fully" sustainable or responsible practice, which is not a particularly realistic or informative concept. Pending more widespread use of the Tool and consideration of the results, we also caution against drawing conclusions from comparisons of scores between different countries, given the varied contexts in which countries operate.

In summary, application of the Tool requires informative, representative, and consistent data inputs. This relies on input from people knowledgeable about the country or jurisdiction in question, with experience with policy, fishery management implementation, and fisheries operations, and good data on essential details like catch and revenue. To be most successful, users of the Tool should also be familiar with the existing standards and certifications referenced in its development, even if none of their fisheries have been through an evaluation using these, nor intend to do so.

The initial aim of the work was to create an internal evaluation tool for WFF to assess the current state and progress of fisheries management and inform its grant-making strategies. As of this writing, the Tool has been put to use in iterative assessment and tracking progress of WFF's priority fisheries in Chile, Peru, Indonesia, Mexico, and the United States. The initially unplanned result of external review and testing of the Tool was the shift from the Foundation's internal strategy, learning and evaluation needs to an accessible and user-friendly, public-facing Tool aimed at a broad range of other interested stakeholders. The frequency of application of the Tool following its launch in October 2020, and the extent of its contribution to the evaluation of sustainable fisheries governance remain to be seen.

The authors invite comments and encourage constructive criticism of the Tool, and look forward to more adaptations and improvements once the Tool is in use more widely.

## Supporting information

**S1 Table. Standards of evidence and examples for assessing whether or not a measure is met under Component 1: Policy.**
(DOCX)

**S2 Table. Standards of evidence and examples for assessing whether or not a measure is met under Component 2: Capacity.**
(DOCX)

**S3 Table. Standards of evidence and examples for assessing whether or not a measure is met under Component 3: Fisheries performance.**
(DOCX)

## Acknowledgments

The authors thank the anonymous reviewers for their insightful suggestions and careful reading of the manuscript. We are thankful to ProsperSpark for their creative and insightful development of the downloadable Tool. We appreciate the contribution of the following for their participation in the peer-review work group, testing, and helping us improve the Tool: Dr. Transform Aqorau, Pacific fisheries export and former Chief Executive Officer of the Parties to the Nauru Agreement; Ian Cartwright, Commissioner, Australian Fisheries Management Authority; Stefan Gelcich, assistant professor, Pontificia Universidad Catolica, Chile; Rebecca Goldburg, Director, Environmental Science, The Pew Charitable Trusts; Rebecca Lent, Executive Secretary, International Whaling Commission; Douglas N. Rader, Chief Oceans Scientist, Environmental Defense Fund; Andrew Rosenberg, Ph.D., Director, Center for Science and Democracy; John Virdin, Director, Ocean and Coastal Policy Program, Nicholas Institute for Environmental Policy Solutions, Duke University; and Santi Roberts, Monterey Bay Aquarium Seafood Watch.

## Author Contributions

**Conceptualization:** Cheri A. Recchia.

**Formal analysis:** Jill H. Swasey, Suzanne Iudicello, Robert Trumble.

**Funding acquisition:** Kara Stevens, Martha Silver, Cheri A. Recchia.

**Investigation:** Jill H. Swasey, Suzanne Iudicello, Kara Stevens, Martha Silver.

**Methodology:** Jill H. Swasey, Suzanne Iudicello, Graeme Parkes, Robert Trumble.

**Project administration:** Jill H. Swasey.

**Supervision:** Graeme Parkes.

**Validation:** Suzanne Iudicello, Graeme Parkes.

**Visualization:** Jill H. Swasey, Kara Stevens, Martha Silver.

**Writing – original draft:** Jill H. Swasey, Suzanne Iudicello, Graeme Parkes.

**Writing – review & editing:** Robert Trumble, Kara Stevens, Martha Silver, Cheri A. Recchia.

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
