## [Decision Letter · Decision Letter 0]

26 Feb 2021

PONE-D-20-40251

The Fisheries Governance Tool: A Practical and Accessible Approach to Evaluating Management Systems

PLOS ONE

Dear Dr. Parkes,

Thank you for submitting your manuscript to PLOS ONE. After careful consideration, we feel that it has merit but does not fully meet PLOS ONE’s publication criteria as it currently stands. Therefore, we invite you to submit a revised version of the manuscript that addresses the points raised during the review process.

The reviewers were complimentary about your manuscript, but it needs an extensive revision to meet the journal's publication criteria. The main issue is the lack of connection with academic literature; thus, there is a need for better linking your approach with the ones published in scientific journals. Also, both reviewers suggested revisions, and I highly advise you to incorporate them into your paper as much as possible, and if it is not possible, please explain why.

We look forward to receiving your revised manuscript.

Kind regards,

Laurentiu Rozylowicz, Ph.D.

Academic Editor

PLOS ONE

Journal Requirements:

2.Thank you for stating the following in the Financial Disclosure section:

"This work was supported through a contract from the Walton Family Foundation to MRAG Americas. WFF staff, as identified in the author list, participated in Funding Acquisition, Investigation, Visualization, Writing – Review & Editing.

Funds were received by MRAG Americas through a consulting agreement. No actual grants were made."

We note that one or more of the authors are employed by a commercial company: MRAG Americas, Inc. and Iudicello Environmental Consulting

3.We note that the grant information you provided in the ‘Funding Information’ and ‘Financial Disclosure’ sections do not match.

Reviewers' comments:

Reviewer's Responses to Questions

**Comments to the Author**

1. Is the manuscript technically sound, and do the data support the conclusions?

Reviewer #1: Yes

Reviewer #2: Partly

2. Has the statistical analysis been performed appropriately and rigorously? 

Reviewer #1: N/A

Reviewer #2: N/A

3. Have the authors made all data underlying the findings in their manuscript fully available?

Reviewer #1: Yes

Reviewer #2: Yes

4. Is the manuscript presented in an intelligible fashion and written in standard English?

Reviewer #1: Yes

Reviewer #2: Yes

5. Review Comments to the Author

Reviewer #1: It is an interesting paper primarily intended to document an evidence-based diagnostic tool namely, the ‘Fisheries Governance Tool’ which can be made use of to evaluate the performance of fisheries in varied regional settings. The authors have provided detailed information on the motivation, theoretical foundation, structure and empirical applicability as well as other necessary particulars with regard to the Tool in a systematic fashion. The Tool, as described in the paper is based on scientific principles of fishery management, and has taken into account the relevant international instruments/guidelines/assessment protocols presently in force. Moreover, a detailed perusal of the workings of the downloadable application clarifies its practical utility beyond doubt. Overall, the paper has promising potential to be published. I am suggesting a few points which may be considered while revising the paper for better clarity.

• In section 2.3.2. scoring, the authors mention that the four scales (basic; adequate; good; better) for each measure are scored at an incremental rate which is quite sensible (even though a bit arbitrary). However, it is not clear whether such differential weighting/scoring scheme is applied across measures within the same component, and across components. If not, how it is expected that different measures contribute with same effectiveness towards the sustainability/efficiency of a fishery?

• Even though the authors discuss about data-deficient fisheries at a few places, they may elaborate on how the Tool can circumvent the problems of small-scale fisheries where adequate systems are not in place to systematically document different aspects of the fishery, as the Tool demands. Here, I would like to refer the specific case of certain island ecosystems where the ecosystem is pristine and the inhabitants take utmost care in maintaining sustainable fishing practices over centuries based on traditional management approaches, but they do not have sufficient data/documentations in place.

• The authors may also elaborate on how non-state legal systems/co-management systems are taken into account in the Tool under different components/indicators/measures.

• Whether the Tool has the potential to be utilized for developing independent eco-certification schemes in the future? – This aspect may be discussed in the concluding section.

Reviewer #2: This paper describes the Fisheries Governance Tool - a new and welcome approach to move to performance assessments which are pointing to ways to enable fisheries management. Overall, it is a clear description and a good introduction to the tool. However, my main problem with this paper to be published in this academic journal lies in the absence of connection to the academic literature. This is already indicated by the few references to the existing body of literature. Although Anderson et al's FPIs are mentioned, there is little contrasting so to understand what has been learnt/taken from Anderson et al, and how the FGT is different. In reading, I felt it is mentioned but the how is not well-explained. I think the paper would benefit from a more specific linking between section 1.1 and 1.2. Doing so, the authors could balance in 1.1 between what is context and what has been taken from the review to develop the FGT. Also, a more specific comparison would clarify the specific characteristics of the FGT, such as why the FGT is adaptable in scale, while the FPIs are not (because one could argue that, in principle, they are too).

It is clear that FGT is more practice-informed than theory-informed, and this is appreciated given the problem statement presented in the introduction. However, given their ambition to publish in this journal, the authors should elaborate more on the underpinning of their tool, for example it would be helpful if they would clarify what theoretical notions are key to their understanding of governance. The three components "policy, capacity and performance" are relevant, yet it is presented being common sense, rather than explained and substantiated. There is perhaps no need to define policy, but what is "capacity" or "performance" to the authors? Even some more attention to the choice of underpinning the tool by the "triple bottom line" would help understanding its position.

In my opinion, the attention to WFF in the introduction is not fitting with the nature of an academic publication. WFF's role should not be understated, but now is central to the problem framing while the relevance of this tool goes beyond its use by WFF (as the authors rightly state). I think the authors would do more justice to their tool if they would put that broader relevance/usability already forward in the introduction. I think the reader would also appreciate a short outline of the paper. The introduction now concludes with a claim about the FGT, but it is unclear whether this is what we will see evidenced in the paper. From my reading, I don't think the authors show "the FGT as a means of consistently assessing a country's progress over time" and how the FGT provides a road map. For example, it does not showcase examples of its use (however, that is not needed for a publication like this one. The goal of presenting the tool, and showing how it fits in with the current toolbox, is already valuable).

I hope these comments are of value to the authors, because when improved, the paper is a much needed addition to show that we need better tools and approaches to help understand how sustainability performance is related to governance processes.

6. PLOS authors have the option to publish the peer review history of their article (what does this mean?). If published, this will include your full peer review and any attached files.

Reviewer #1: **Yes: **Shinoj Parappurathu

Reviewer #2: No

---

## [Author Response · Author response to Decision Letter 0]

24 May 2021

Comments to the Author have been abstracted from the full text. 

Responses include line numbers that refer to the revised manuscript without tracked changes

Comment:

Provide a short outline of the paper.

Response:

Added text at the end of the Introduction– lines 81 to 86

Comment:

The attention to WFF in the introduction is not fitting with the nature of an academic publication. WFF's role should not be understated, but the relevance of this tool goes beyond its use by WFF (as the authors rightly state).

Response: 

We have edited the Introduction to focus more on the FGT than WFF, and also the motivation for preparing the manuscript for publication (lines 75-80).

Comment:

Put that broader relevance/usability forward in the introduction.

Response:

The edits to the Introduction in response to the previous comments also address this comment, e.g. lines 75-80 and 81-86. Also there is text in the new section 2 (previously 1.1/1.2) that better explains the relevance of the tool in the context of other tools.

Comment:

Better explain the tool:

Elaborate more on the underpinning of their tool, for example it would be helpful if they would clarify what theoretical notions are key to their understanding of governance.

The three components "policy, capacity and performance" are relevant, yet it is presented being common sense, rather than explained and substantiated. There is perhaps no need to define policy, but what is "capacity" or "performance" to the authors? Even some more attention to the choice of underpinning the tool by the "triple bottom line" would help understanding its position. 

Response:

The explanation of three components of policy, capacity and performance is expanded in new text in section 3.1 (Outline of the FGT) (lines 194-212). 

The introduction now concludes with a claim about the FGT, but it is unclear whether this is what we will see evidenced in the paper. From my reading, I don't think the authors show "the FGT as a means of consistently assessing a country's progress over time" and how the FGT provides a road map. This claim is deleted. While true, the claim relies on the consistent use of the tool over time, and is not a characteristic of the tool itself.

Comment:

Absence of connection to the academic literature :

The paper would benefit from a more specific linking between section 1.1 and 1.2. Doing so, the authors could balance in 1.1 between what is context and what has been taken from the review to develop the FGT. Also, a more specific comparison would clarify the specific characteristics of the FGT, such as why the FGT is adaptable in scale, while the FPIs are not (because one could argue that, in principle, they are too).

Although Anderson et al's FPIs are mentioned, there is little contrasting so to understand what has been learnt/taken from Anderson et al, and how the FGT is different. In reading, I felt it is mentioned but the how is not well-explained. 

Response:

Sections 1.1 and 1.2 have been amalgamated in a new section 2. The new section presents a more evenly weighted description of existing evaluation mechanisms and concludes with a more clear explanation of the design criteria for the FGT leading to the decision to build a novel tool. 

Comment:

In section 2.3.2. scoring, the authors mention that the four scales (basic; adequate; good; better) for each measure are scored at an incremental rate which is quite sensible (even though a bit arbitrary). However, it is not clear whether such differential weighting/scoring scheme is applied across measures within the same component, and across components. If not, how it is expected that different measures contribute with same effectiveness towards the sustainability/efficiency of a fishery?

Response:

New text has been added to more fully explain the scoring and it’s limitations in this first iteration of the Tool (lines 366-397)

Comment: 

Even though the authors discuss about data-deficient fisheries at a few places, they may elaborate on how the Tool can circumvent the problems of small-scale fisheries where adequate systems are not in place to systematically document different aspects of the fishery, as the Tool demands. Here, I would like to refer the specific case of certain island ecosystems where the ecosystem is pristine and the inhabitants take utmost care in maintaining sustainable fishing practices over centuries based on traditional management approaches, but they do not have sufficient data / documentations in place. 

Response:

We have added new text into Section 4 Discussion regarding use of the Tool with small scale and data deficient fisheries (lines 503-524).

Comment:

The authors may also elaborate on how non-state legal systems/co-management systems are taken into account in the Tool under different components / indicators / measures.

Response:

The new text added into Section 4 Discussion also addresses this point (lines 503-524).

Comment:

Whether the Tool has the potential to be utilized for developing independent eco-certification schemes in the future? – This aspect may be discussed in the concluding section. 

Response:

We have declined to comment on this specifically, believing it would be premature. The Tool could be used in a variety of ways, some of which have not been identified, and/or thought of as yet, and may only become apparent after a period of use in a variety of circumstances. The Tool is specifically set up without an externally set standard or benchmark, leaving it up to the user to set their own goals. The FGT is not intended to be used as a certification scheme, but neither do we feel we should explicitly state this in the manuscript, given the debate such an explicit “denial” might provoke. We consider such a debate would detract from the central purpose of the Tool at this early stage. Nevertheless, we would be potentially interested to revisit this in the future.

---

## [Decision Letter · Decision Letter 1]

14 Jun 2021

The Fisheries Governance Tool: A Practical and Accessible Approach to Evaluating Management Systems

PONE-D-20-40251R1

Dear Dr. Parkes,

We’re pleased to inform you that your manuscript has been judged scientifically suitable for publication and will be formally accepted for publication once it meets all outstanding technical requirements.

Kind regards,

Laurentiu Rozylowicz, Ph.D.

Academic Editor

PLOS ONE

Additional Editor Comments (optional):

Reviewers' comments:

Reviewer's Responses to Questions

**Comments to the Author**

1. If the authors have adequately addressed your comments raised in a previous round of review and you feel that this manuscript is now acceptable for publication, you may indicate that here to bypass the “Comments to the Author” section, enter your conflict of interest statement in the “Confidential to Editor” section, and submit your "Accept" recommendation.

Reviewer #1: All comments have been addressed

2. Is the manuscript technically sound, and do the data support the conclusions?

Reviewer #1: Yes

3. Has the statistical analysis been performed appropriately and rigorously? 

Reviewer #1: N/A

4. Have the authors made all data underlying the findings in their manuscript fully available?

Reviewer #1: Yes

5. Is the manuscript presented in an intelligible fashion and written in standard English?

Reviewer #1: Yes

6. Review Comments to the Author

Reviewer #1: The authors have satisfactorily addressed (or explained) my comments, thus improving the manuscript adequately. The revised manuscript may be accepted for publication.

7. PLOS authors have the option to publish the peer review history of their article (what does this mean?). If published, this will include your full peer review and any attached files.

Reviewer #1: **Yes: **Shinoj Parappurathu

---

## [Editor Report · Acceptance letter]

21 Jun 2021

PONE-D-20-40251R1 

The Fisheries Governance Tool: A Practical and Accessible Approach to Evaluating Management Systems 

Dear Dr. Parkes:

I'm pleased to inform you that your manuscript has been deemed suitable for publication in PLOS ONE. Congratulations! Your manuscript is now with our production department. 

Kind regards, 

on behalf of

Dr. Laurentiu Rozylowicz 

Academic Editor

PLOS ONE